# Serum Exosomal microRNA-21, 222 and 124-3p as Noninvasive Predictive Biomarkers in Newly Diagnosed High-Grade Gliomas: A Prospective Study

**DOI:** 10.3390/cancers13123006

**Published:** 2021-06-15

**Authors:** Debora Olioso, Mario Caccese, Alessandra Santangelo, Giuseppe Lippi, Vittorina Zagonel, Giulio Cabrini, Giuseppe Lombardi, Maria Cristina Dechecchi

**Affiliations:** 1Department of Neurosciences, Biomedicine and Movement, University of Verona, 37126 Verona, Italy; debora.olioso@univr.it (D.O.); alessandra.santangelo@univr.it (A.S.); giuseppe.lippi@univr.it (G.L.); giulio.cabrini@univr.it (G.C.); 2Department of Oncology Oncology 1, Veneto Institute of Oncology IOV-IRCSS, 35128 Padova, Italy; mario.caccese@iov.veneto.it (M.C.); vittorina.zagonel@iov.veneto.it (V.Z.); giuseppe.lombardi@iov.veneto.it (G.L.)

**Keywords:** liquid biopsy, high-grade glioma, circulating miRNAs, exosomes

## Abstract

**Simple Summary:**

Diagnosis of relapse during post-surgery monitoring of patients with High-grade glioma is often challenging. This study was aimed to identify circulating biomarkers providing information about response to therapy and early diagnosis of progression during follow-up of these patients. Our findings, showing upregulation of exosomal miRNAs associated with relapse represent a proof of principle that these biomarkers may be clinically useful tools for diagnosing and monitoring of gliomas.

**Abstract:**

*Background:* High-grade gliomas (HGG) are malignant brain tumors associated with frequent recurrent disease. Clinical management of HGG patients is currently devoid of blood biomarkers for early diagnosis, monitoring therapeutic effects and predicting recurrence. Different circulating miRNAs, both free and associated with exosomes, are described in patients with HGG. We previously identified miR-21, miR-222 and miR-124-3p purified from serum exosomes as molecular signature to help pre-operative clinical diagnosis and grading of gliomas. The aim of the present study was to verify this signature as a tool to assess the effect of treatment and for the early identification of progression in newly diagnosed HGG patients. *Material and Methods:* Major inclusion criteria were newly diagnosed, histologically confirmed HGG patients, no prior chemotherapy, ECOG PS 0-2 and patients scheduled for radiochemotherapy with temozolomide as first-line treatment after surgery. RANO criteria were used for response assessment. Serum was collected at baseline and subsequently at each neuroradiological assessment. mir-21, -222 and -124-3p expression in serum exosomes was measured in all samples. *Results**:* A total number of 57 patients were enrolled; 41 were male, 52 with glioblastoma and 5 with anaplastic astrocytoma; 18 received radical surgery. HGG patients with higher exosomal miRNA expression displayed a statistically significant lower progression-free survival and overall survival. Increased expression of miR-21, -222 and -124-3p during post-operative follow-up was associated with HGG progression. *Conclusions:* These data indicate that miR-21, -222 and -124-3p in serum exosomes may be useful molecular biomarkers for complementing clinical evaluation of early tumor progression during post-surgical therapy in patients with HGG.

## 1. Introduction

Gliomas are malignant tumors of the central nervous system (CNS) of glial origin. Glioblastoma (GBM) is the most common and aggressive brain tumor, still characterized by a dismal prognosis [1]. Standard treatment consists of maximal safe surgical resection followed by combined radiochemotherapy with temozolomide (TMZ) according to Stupp’s protocol [2]. Despite treatment, GBM recurrence is commonplace.

Extensive neovascularization is a common finding of progression, revealed by magnetic resonance imaging (MRI) as an increase of tumor mass. However, increased vessel permeability causing edema is observed at MRI after radiochemotherapy in approximately 20–30% of all patients. Thus, a pseudoprogression can mimic true progression. Since no imaging techniques are currently available to reliably distinguish between recurrence and pseudoprogression, clinically useful biomarkers capable of indicating tumor progression are an unmet need. In this respect, liquid biopsy (i.e., noninvasive analysis of tumor-derived material present in bodily fluids) represents a promising tool both for diagnosing gliomas and supporting clinicians during longitudinal disease monitoring [3]. Different tumor-derived material, such as proteins, circulating tumor DNA, mRNA, microRNA (miRNA), extracellular vesicles (EVs) and tumor-educated platelets can be detected in the blood and cerebrospinal fluid (CSF) of patients with gliomas [3,4]. Despite mutations in circulating tumor-derived DNA, changes in mRNA and miRNA expression and even alterations of protein levels having been described in blood and CSF, none of these biomarkers has been validated for the diagnosis and prognosis of gliomas.

EVs are small lipid-membrane particles released from all cells under different conditions to mediate intercellular communication; they regulate many different physiological and pathological processes. They can be divided into two groups: microvesicles and exosomes. Recent data support circulating exosomes as a promising source for biomarker discovery, as they can be collected by noninvasive procedures, their half-life in vivo is short and their content is protected from degradation [5,6].

Potential support for glioma diagnosis and prognosis might come from miRNAs, small non-coding RNAs, which have a key role in regulating gene expression, mainly interacting with target mRNAs and suppressing transcription. Different circulating miRNAs, both free and associated with exosomes, have been described in blood and CSF of patients with brain tumors, in particular high-grade gliomas (HGG) (for ref. see [7]). Regarding the importance of miRNAs in gliomas, they are involved in glioma genesis as oncogenes or tumor suppressors, promote immune evasion [8] and alter blood–brain-barrier permeability [9]. Therefore, they would seem ideal diagnostic and prognostic biomarkers and could even be considered novel molecular targets for innovative therapies for HGG.

Numerous studies documented the clinical relevance of circulating miRNAs for CNS tumor diagnosis, prognosis and response to therapy (for a review, see [10]). Among these miRNAs, miR-21 has been confirmed as the most powerful biomarker of gliomas. The upregulation of circulating miR-221/222 may be used for diagnosis and the prediction of outcomes in patients with GBM [11]. Dysregulation of miR-124-3p has been found in patients with brain tumors [12].

We previously identified a signature of three miRNAs, namely miR-21, miR-222 and miR-124-3p, in serum exosomes for the pre-operative diagnosis and grading of gliomas [13]. The pre-operative expression of miR-21, miR-222 and miR-124-3p was significantly higher in patients with HGG than in healthy controls or in patients with low-grade gliomas (LGG). Interestingly, the combined expression of mir-21, mir-222 and miR-124-3p sharply decreased after the surgical ablation of tumor mass in patients with HGG, thus outlining their potential application during longitudinal disease monitoring.

The aim of the present study was to validate this circulating exosomal miRNA signature as a tool to evaluate the response to treatment and for the early identification of progression in patients with HGG.

To this aim, we enrolled 57 patients after surgical removal of HGG, before starting radiotherapy (RT) and TMZ cycles. Patients were monitored during treatment by blood sampling and MRI performed at each time point. Treatment response was evaluated according to RANO criteria. Patients were considered *“Progressed”* when progression occurred during the follow-up or “*Stable*” if they had no clinical evidence of recurrence up to the end of the study.

Data obtained indicate that miR-21, miR-222 and miR-124-3p in serum exosomes are clinically useful biomarkers to monitor progression during Stupp’s protocol in patients with HGG.

## 2. Results

### 2.1. Study Design

Patients were enrolled about 4 weeks after surgery (after the complete healing of the surgical wound). As shown in Figure 1, for the scheduled visit before the start of the concomitant chemoradiotherapy treatment, patients performed tumor assessment with brain MRI, and a blood sample was collected for the evaluation of exosomal miRNAs (T1). The second blood sample was obtained at the end of the concomitant chemoradiotherapy treatment and before the start of maintenance therapy with temozolomide (T2), with brain MRI as radiological tumor assessment. Subsequently, blood samples were collected every two cycles of maintenance chemotherapy with temozolomide (T3, T4, T5) concurrently with neuroradiological assessment with brain MRI.

In case of disease progression/relapse, the last sample was taken according to the established time point, and then the patient was excluded from subsequent evaluations.

### 2.2. Patients

A total of 57 patients were enrolled. Clinical data of these patients are reported in Table 1.

### 2.3. Association of Circulating miRNA Expression with PFS or OS

We previously demonstrated that miR-21, miR-222 and miR-124-3p expression in serum exosomes is associated with glioma grade [13], thus suggesting that high pre-operative miRNA expression may predict worse prognosis in patients with HGG. In order to ascertain the prognostic significance of these exosomal miRNAs, they were analyzed by Kaplan–Meier and log-rank models. At the time of analysis, 36 patients had progression, and 21 patients were assessed as stable. For all patients, the median progression-free survival (PFS) was 8.5 (95% CI, 1–41) months and median overall survival (OS) was 15.5 (95% CI, 3–43) months.

As no reference or cut-off value is available for serum exosomal miRNAs, we arbitrarily chose the median value, as in previous experience [14]. Patients were categorized into 2 groups according to miRNA expression levels, measured at each time point, as above median expression (high) or below (low), and a comparison with PFS and OS was performed. Results are summarized in Table 2. At time points T1 and T2, no association of exosomal miRNA expression with PFS or OS could be observed. As assessed by the log-rank test, the median PFS was significantly (*p* < 0.006) increased from 7.3 to 10.2 months in patients with lower miR-21 expression after RT and first TMZ cycles (T3); additionally, median OS (*p* > 0.048) was prolonged from 16.9 to 23.5 months. Exosomal miR-21 levels found at T4 confirmed the association between higher expression and lower median PFS (7.9 versus 11.0 months, *p* < 0.032) and OS (14.6 versus 26.2, *p* < 0.032). Interestingly, increased median PFS of 2 months (8.4 versus 10.4 months) and OS of about 9 months (14.6 versus 23.9 months) could be observed in patients with low miR-124-3p expression (log-rank *p* value of 0.048 and 0.031, respectively). A significant association (*p* < 0.047) between lower miR-222 expression and prolonged median OS of about 10 months (16.9 versus 26.2 months) was also found at this time point. At the end of the study (T5), we observed a significant association of all miRNAs with PFS and OS (see values reported in Table 2).

Although we failed to find significant differences in PFS or OS in patients with higher miRNA expression in serum exosomes obtained before starting Stupp’s protocol (T1) or after RT and first TMZ cycle (T2), our results reveal that HGG patients with higher exosomal miRNA expression were characterized by poorer PFS and OS during later stages of follow-up. Interestingly, Kaplan–Meier analysis indicates that miR-21 was the earliest of these three biomarkers in predicting differences in PFS and OS.

### 2.4. Exosomal miRNA Expression during Longitudinal Follow-Up

In order to evaluate whether changes in exosomal miRNA expression during follow-up can be useful biomarkers to monitor the response to therapy and predict disease progression, we carried out a comparison between Stable and Progressed patients at each time point. As shown in Figure 2, no differences were found in miR-21 (A) and miR-222 (B) expression between the two groups of patients at both T1 and T2. Unexpectedly, median miR-124-3p expression was significantly lower in Progressed patients compared to Stable at the beginning of the study, before starting Stupp’s protocol (T1), whilst no differences could be observed after RT and first cycle of TMZ (T2). The median expression of each miRNA was increased in Progressed and decreased in Stable patients at T3, though this difference was statistically significant only for miR-21 (A). The median miRNA expression further increased in Progressed and decreased in Stable patients at the later time points T4 and T5. Based on these data, the reduction of exosomal miRNA expression observed during the monitoring of Stupp’s protocol could suggest an effective response to therapy, whilst increased expression mirrors disease progression in HGG.

The trend over time of median expression of each miRNA in the two groups of patients is shown in Figure 2D–F. For all three miRNAs, a decrease of median expression along the treatment could be observed, starting from T2 in Stable patients. As shown in Figure 2D, miR-21 expression was significantly lower in Stable compared to Progressed patients, from T3 to T5, whilst miR-222 (Figure 2E) and miR-124-3p (Figure 2F) expression appeared significantly decreased only at T5 (miR-222) and at T4 and T5 (miR-124-3p), respectively.

### 2.5. Exosomal miRNAs Predict Progression in HGG Patients

The ability of each miRNA to predict progression was also assessed by means of ROC curve analyses, as shown in Figure 3A–C. Patients were categorized based on miRNA expression levels by comparing Progressed versus Stable patients as internal control, at each time point. The ROC curve and the area under the ROC curve (AUC) confirmed that at both T1 and T2, miR-21 (A) and miR-222 (B) did not efficiently discriminate Stable from Progressed patients, as indicated by low AUCs (T1: miR-21 AUC: 0.52; miR-222 AUC: 0.51), (T2: miR-21 AUC: 0.53; miR-222 AUC: 0.58). In agreement with data shown in Figure 2C, low miR-124-3p expression measured before starting therapy could predict poorer prognosis (AUC: 0.71 and *p* value of 0.04) (C). At T3, exosomal miR-21 was the earliest biomarker to predict response to therapy (AUC: 0.78 and *p* value of 0.01) (A). At the later stages of the study (i.e., T4 and T5), all these three miRNAs very efficiently discriminated Progressed from Stable patients. These data further support the predictive potential of serum exosomal miR-21, miR-222 and miR-124-3p in the therapeutic monitoring of HGG patients.

### 2.6. Exosomal miRNA Expression in Clinical Management of HGG Patients

To ascertain the potential role of these biomarkers in the clinical management of HGG patients, we measured the differences between expression levels at T3 and T4 and those found at T1 in each patient separately, in order to monitor the variation of expression during the follow-up and calculate the percentage of patients belonging to the *Stable* and *Progressed* groups who show increased expression levels of all the three miRNAs, two, one or none. As reported in Figure 4, the analysis of patients at T3 shows increased levels of expression of miRNAs preferentially in those who later underwent progression at T5. Increase of 1, 2 or 3 miRNAs is associated with higher percentages to *Progressed* patients (64, 78 or 89%, respectively), meaning that the increase of expression of 2 or 3 miRNAs is strongly in favor of a trend of relapse at least at time T5 (33 weeks after surgery). Even more clearly, at T4 (25 weeks post-surgery), the presence of 2 or 3 miRNAs increased compared to the expression level immediately post-surgery (T1), which is 100% associated with relapse at time T5. These results suggest that increased expression of at least 2 miRNAs at time T3 and T4 compared to post-surgery levels could be considered an early biomarker of disease progression, even before radiological evidence.

## 3. Discussion

One of the main goals of measuring circulating biomarkers is to provide accurate information about response to therapy during disease monitoring in HGG patients. The key finding of the present study is that increased expression of serum exosomal miR-21, miR-222 and miR-124-3p during post-operative follow-up is associated with HGG progression. Actually, patients with high miRNA expression after RT and three cycles of TMZ had significantly shorter PFS and worse OS. Independently of levels of miRNA expression, PFS and OS were 8.5 and 15.5 months, respectively, whilst prolonged OS was found in the group of patients with low expression when patients were stratified based on miRNA expression levels, starting from T3 (17 weeks after surgery). Although no correlation could be found between exosomal miRNA expression and PFS or OS at the beginning of the study or after RT and first cycle of TMZ, data reported here show that exosomal miRNAs may predict response to therapy during follow-up in HGG patients. Circulating exosomal miRNA expression found before starting Stupp’s protocol may depend on many different factors, such as pre-operative levels or extent of surgical ablation. In this study, no information on pre-operative miRNA expression was available, so we cannot assess the prognostic potential of the signature before surgery in this patient cohort. The unexpected lower expression of miR-124-3p found in Progressed patients at T1 is difficult to explain and needs to be further investigated in a larger group of patients. Similarly to miR-21 and miR-222, the expression of miR-124-3p increased during follow-up in Progressed patients, consistent with association between higher miRNA expression and disease progression. However, independently of the levels found at the beginning of the study, reduction of miRNA expression during follow-up reveals effective response to treatment. Among the three miRNAs, miR-21 was the earliest predictive biomarker. MiR-21 was one of the first mammalian miRNAs described to be upregulated in many different types of cancer, including gliomas [15,16,17]. It was proposed as a diagnostic, prognostic and predictive cancer biomarker [16]. Importantly, modulation of miR-21 expression seems to play a broad role in sensitivity to chemotherapy in cancer cells. Notably, increased miR-21 expression has been linked to resistance to platinum-based chemotherapy [18], as well as to gemcitabine [19]. Furthermore, increased serum miR-21 levels were found to predict worse response to chemotherapy with trastuzumab [20]. miR-21 has also been shown to mediate resistance of GBM cells to both RT and TMZ through all its target genes involved in apoptosis, invasion, proliferation, tumor growth and chemoresistance, thus suggesting that lower miR-21 expression during treatment might implicate effective response to therapy [21,22]. From the data shown in the present study, it can be speculated that increasing circulating exosomal miR-21 expression may reflect escape from first-line treatment. Monitoring patient response to therapy is particularly challenging in HGG. The longitudinal assessment conducted in our study during Stupp’s treatment was aimed at evaluating whether changes in exosomal miRNA levels may be related to tumor regrowth in post-surgery follow-up. Since cut-off values for each miRNA could not be established, differences between levels of miRNA expression at each time point and those found at the beginning of the study (T1) were taken into account. Our data demonstrate that an increased expression of all three miRNAs is strongly associated with glioma progression, starting from T3, occurring about 17 weeks after surgery. Importantly, only 30% of Progressed patients already had clinical evidence of progression at T3, whilst increased exosomal miRNA levels were early biomarkers of progression in all the other Progressed patients. To the best of our knowledge, no dynamic changes in exosomal miRNAs as a function of glioma evolution has been published to date. Thus, our study reports for the first time increased exosomal miRNA expression, associated with disease progression. In summary, these findings support the use of exosomal miR-21, miR-222 and miR-124-3p expression as complementary molecular biomarkers for the clinical evaluation of tumor relapse during post-surgical longitudinal monitoring of HGG patients.

## 4. Materials and Methods

### 4.1. Patients and Sample Collection

Ethics approval was obtained (CESC IOV 2016/83), and written informed consent was required from all patients. The study was carried out in accordance with the Declaration of Helsinki and Good Clinical Practice guidelines.

Clotted samples were centrifuged at room temperature at 1500× *g* for 15 min and serum aliquoted and stored at −80 °C within 2 h from collection.

*MGMT* promoter methylation status was analyzed by pyrosequencing using a commercial kit (MGMT plus, Diatech Pharmacogenetics, Milano, Italy) on a PyroMark Q96 ID system equipped with PyroMark CpG (Qiagen, Redwood City, CA, USA) software. IDH status was investigated by Sanger sequencing.

### 4.2. Purification of Exosomes

This was performed as described [13] within 3 months of sample collection. Exosomes were purified from 500 µL of serum by ExoQuick-TM (System Biosciences Inc., Palo Alto, CA, USA) following the manufacturer’s instructions. Briefly, serum samples were thawed on ice and centrifuged at 3000× *g* for 15 min to remove cell debris, then incubated with one-fourth volume of ExoQuick solution for 1 h on ice in vertical position. After centrifugation at 1500× *g* for 30 min exosomes were collected in 250 µL of nuclease-free water (Invitrogen, Waltham, MA, USA).

### 4.3. RNA Isolation from Exosomes

After purification, exosomes were immediately processed for miRNA isolation as previously described [13]. To increase miRNA yield, samples were lysed with Trizol-LS (ThermoFisher Scientific, Waltham, MA, USA) with the addition of miRNeasy Serum/Plasma Spike-in control as an RNA carrier (Qiagen, Redwood City, CA, USA), and RNA was purified using miRNeasy serum/plasma kit (Qiagen) according to the manufacturer’s protocol. RNA was stored at −80 °C until use.

### 4.4. miRNA Quantification by Real-Time qPCR

miRNA quantification was carried out using real time RT qPCR and TaqMan technology as reported [13]. A standard volume of 5 µL total RNA was added in the reverse transcription reaction, conducted at a final volume of 15 µL for 30 min at 16°, 30 min at 42 °C, 5 min at 85 °C and held at 4 °C using a high-capacity cDNA reverse transcription kit (ThermoFisher Scientific, Waltham, MA, USA). The RT was performed with specific primers for each miRNA and U6 snRNA endogenous control (TaqMan microRNA assays, ThermoFisher Scientific, Waltham, MA, USA; hsa-miR-21 assay ID 002438, hsa-miR-124a assay ID 001182, hsa-miR-222 assay ID 002776 and U6 snRNA assay ID 001973). Finally, 5 µL of 1:2 diluted cDNA was used in RT qPCR reaction performed in duplicate with Fast Advanced Master Mix on a 7900HT Fast Real Time PCR System (Applied Biosystem, Waltham, MA, USA) following the manufacturer’s fast amplification protocol. The expression of each microRNA was normalized to U6 snRNA with −DCq formula (−DCq = mean Cq miRNA − mean Cq U6 snRNA).

### 4.5. Statistical Analyses

Patients were considered Progressed if progression occurred during the follow-up or *Stable* without any clinical evidence of recurrence up to the end of the study.

Differences in miRNA expression levels between Stable and Progressed patients were analyzed with non-parametric *U* Mann–Whitney test. Variations in miRNA expression levels at different time points were evaluated by ANOVA. Receiver operating characteristics (ROC) curves analyses were performed comparing miRNA expression as a continuous predictor in Progressed versus Stable patients by using Graphpad Prism version 8.0. The average area under the ROC curves (AUC) and their 95% confidence intervals with statistical significance are computed using the Bayesian bootstrap technique for continuous prognostic variables based on independent observations in Progressed and Stable patients who represent the control group.

Progression-free survival (PFS) was defined as the time from the start of treatment to the progression of tumor or death; overall survival (OS) was defined as the time from the start of treatment until death of any cause. miRNA median expression levels were arbitrarily chosen for dividing patients into two groups: “High” or “Low” in reference to the median value. PFS and OS in the two subgroups were evaluated using the Kaplan–Meier survival curves, and the log-rank test was used for group comparisons. Patients without PFS or OS were censored at the last assessment.

All the statistics were carried out with GraphPad Prism as specified in figure legends. Statistical significance was declared at 2-tailed *p* < 0.05 for all comparisons.

## 5. Conclusions

Monitoring treatment effectiveness and predicting recurrence in patients with HGG is still an unmet need. Our study provides evidence that assessment of miR-21, -222 and -124-3p expression in serum exosomes could complement the clinical evaluation of early tumor progression during post-surgical therapy in patients with HGG.

## Figures and Tables

**Figure 1 cancers-13-03006-f001:**
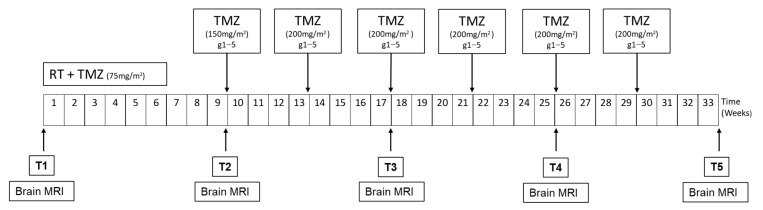
Study design. **RT**, radiotherapy cycle; **TMZ**, temozolomide; **MRI**, magnetic resonance. Blood sampling was done at time points T1, T2, T3, T4 and T5. Duration of the study is expressed in weeks.

**Figure 2 cancers-13-03006-f002:**
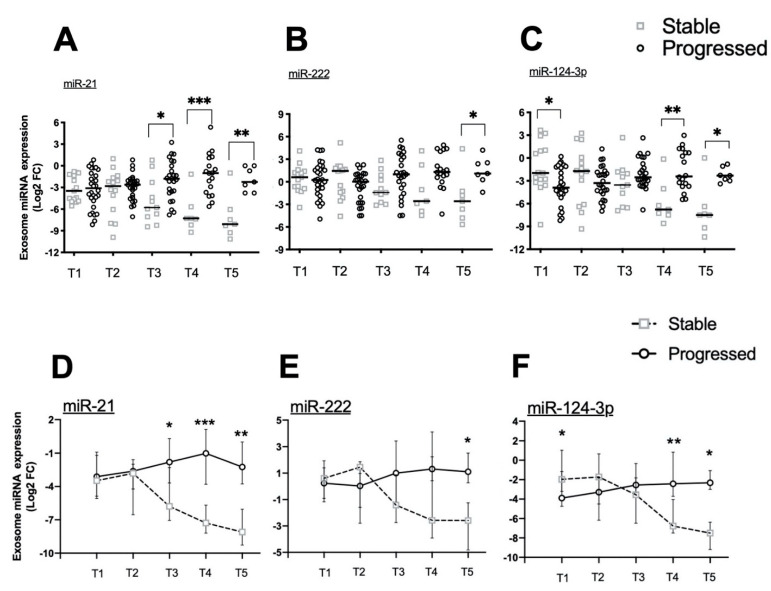
miRNA expression in serum exosomes at different time points in Stable and Progressed patients. miR-21 (**A**), -222 (**B**) and -124-3p (**C**) expression in serum exosomes was measured by real-time qPCR with TaqMan probes and normalized to U6 snRNA. Data are expressed as Log2FC. Comparisons between Stable and Progressed patients at each time point were analyzed with non-parametric *U* Mann–Whitney test. Differences were considered statistically significant as *p* < 0.05 (*), *p* < 0.01 (**), and *p* < 0.001 (***). Median miR-21 (**D**), -222 (**E**) and -124-3p (**F**) expression level in serum exosomes of Stable and Progressed patients was reported at each time point. Variations of miRNA expression levels at different time points were evaluated by ANOVA. Differences were considered statistically significant as *p* < 0.05 (*), *p* < 0.01 (**), and *p* < 0.001 (***).

**Figure 3 cancers-13-03006-f003:**
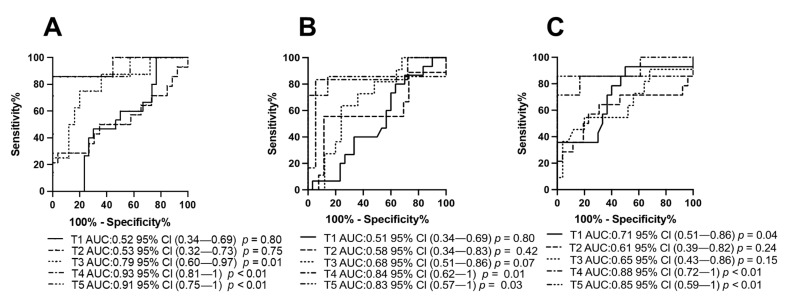
Predictive potential of miRNA expression in serum exosomes at different time points in Stable and Progressed patients. ROC curves showing sensitivity and specificity for miR-21 (**A**); -222 (**B**) and -124-3p (**C**) expression. AUC is area under the curve.

**Figure 4 cancers-13-03006-f004:**
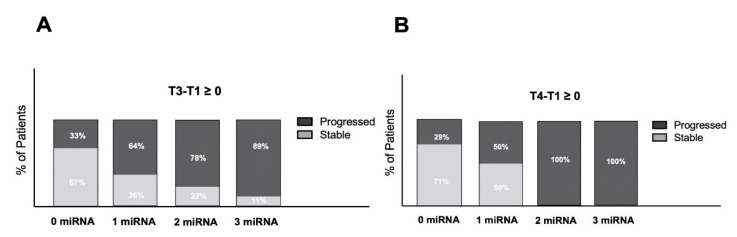
Percentage of patients showing increased expression levels of miRNAs during the follow up. Differences between expression at T3 (17 weeks post-surgery) (**A**) and T4 (25 weeks post-surgery) (**B**) and those at T1 in patients showing *Stable* or *Progressed* disease at T5 (33 weeks post-surgery) were calculated, and the percentage of patients with increased levels of none, 1, 2 or 3 miRNAs is reported.

**Table 1 cancers-13-03006-t001:** Patient characteristics.

Patient Characteristics	N (%)
Total	57
GENDER	
*Male* *Female*	41 (72) 16 (28)
Median Age (range) HISTOLOGY *Glioblastoma* *Anaplastic Astrocytoma*	63 (23–88) 52 (91) 5 (9)
EXTENT OF RESECTION	
*Radical* *Non-Radical* *Unknown*	18 (32) 39 (68) 1 (2)
ECOG PS	
*0–1* *2*	45 (79) 12 (21)
MGMT	
*Methylated* *Unmethylated* *Unknown*	26 (46) 16 (28) 15 (26)
IDH 1/2	
*Wildtype* *Mutated* *Unknown*	49 (86) 3 (5) 5 (9)

**Table 2 cancers-13-03006-t002:** Kaplan–Meier analysis.

		PFS Ratio				OS Ratio			
miRNA	Expression	Median (Months)	High/Low	95% CI	Log-Rank *p*	Median (Months)	High/Low	95% CI	Log-Rank *p*
**T1**									
21	high low	9.1 7.9	1.1	(0.24–1.49)	0.64	17.8 13.1	1.4	(0.75–2.46)	0.51
222	high low	9.1 6.1	1.5	(0.13–1.43)	0.16	14.6 16.2	0.9	(0.46–1.74)	0.24
124-3p	high low	8.6 8.3	1.0	(0.59–1.81)	0.31	17.3 15.6	1.1	(0.62–2.0)	0.77
**T2**									
21	high low	9.1 7.9	1.1	(0.61–2.13)	0.76	17.8 12.9	1.4	(0.71–2.68)	0.96
222	high low	8.6 8.7	1.0	(0.53–1.83)	0.71	14.6 16.2	0.9	(0.46–1.74)	0.24
124-3p	high low	8.8 8.3	1.1	(0.57–1.97)	0.68	17.3 13.1	1.4	(0.70–2.63)	0.68
**T3**									
21	high low	7.3 10.2	0.7	(0.34–1.41)	**0.006**	16.9 23.5	0.7	(0.33–1.51)	**0.048**
222	high low	8.4 9.4	0.9	(0.44–1.78)	0.18	16.9 22.1	0.8	(0.36–1.61)	0.13
124-3p	high low	8.1 8.8	0.9	(0.43–1.85)	0.87	17.8 18.8	0.9	(0.45–1.99)	0.59
**T4**									
21	high low	7.9 11.0	0.7	(0.31–1.68)	**0.032**	14.6 26.2	0.6	(0.21–1.40)	**0.043**
222	high low	9.5 9.7	1.0	(0.42–2.27)	0.29	16.9 26.2	0.6	(0.25–1.83)	**0.047**
124-3p	high low	8.4 10.4	0.8	(0.35–1.82)	**0.048**	14.6 23.9	0.7	(0.24–1.49)	**0.031**
**T5**									
21	high low	10.2 23.1	0.4	(0.13–1.43)	**0.017**	21.3 Und *	Und *	Und *	**0.032**
222	high low	10.2 23.1	0.4	(0.13–1.38)	**0.003**	17.3 35.3	0.5	(0.13–1.74)	**0.047**
124-3p	high low	10.2 23.2	0.4	(0.13–1.38)	**0.022**	21.0 Und *	Und *	Und *	**0.022**

Median PFS and OS (in months) in HGG during follow-up. T1 up to T5 are the time points of blood sampling as described in the study design. Kaplan–Meier survival analyses were tested for statistical significance with a log-rank test reported with median PFS and OS ratio (high/low) and 95% CI (confidence interval). *p* values are marked at <0.05, <0.01 and <0.001 levels. * undetectable.

## Data Availability

The data were not publicly archived.

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
