# Peer review of "Serum Exosomal microRNA-21, 222 and 124-3p as Noninvasive Predictive Biomarkers in Newly Diagnosed High-Grade Gliomas: A Prospective Study"

_cancers, 2021, doi:10.3390/cancers13123006_

Round 1
Reviewer 1 Report
I have some questions and suggestions for authors:
- Since the authors plainly wrote and reported in Table 1 that cohort comprises 41 males and 16 females, did they find any difference related to gender? or to any of the other characteristics listed in Table 1 (histology, ECOG PS, gene status)?
- I recommend the authors to check two points: i) lines 104-106 and lines 107-108 say the same thing; ii) lines 108-110 are exactly the same as lines 306-308 in Methods section. Moreover, I suggest removing Paragraph 4.5 “Kaplan-Meier analysis” in Methods section and adding it inside of the following Paragraph 4.6 “Statistical analyses”.
- To improve readability of Table 2, I suggest aligning all the titles for time points, from T2 to T5, on the left.
- Figure 1 and Figure 2 a-b-c are a little redundant in my opinion. I suggest maintaining only one. In case you prefer to keep both, I invite the authors to combine them in a single multi-panel figure (new Figure 1), but separate from ROC curve (Figure 2).
- Finally, minor spell check is recommended. I found some typos throughout the draft, for example: [line 27] Mir, [line 53] such us, [line 119] with low miR-124-3p expression with, [line 295] 16° (missing C).
- I really appreciated the acknowledgment to patients participating to the study. Good!
Author Response
MS “Serum exosomal microRNA-21, 222 and 124-3p as noninvasive predictive biomarkers in newly diagnosed high-grade gliomas: a prospective study”
Reply to Reviewer #1 (R1)
We thank the Reviewer for her/his very careful revision and useful suggestions to improve the overall presentation of the results. The manuscript has been revised according to the comments of the Reviewer. We hope we have been able to implement the quality of the manuscript.
Question 1.
Since the authors plainly wrote and reported in Table 1 that cohort comprises 41 males and 16 females, did they find any difference related to gender? or to any of the other characteristics listed in Table 1 (histology, ECOG PS, gene status)?
Answer 1.
In our population, no significant difference was found about gender (male vs female), histology (GBM vs AA), ECOG PS (0-1 vs 2), MGMT (met vs unmet) and IDH (wild-type vs mutated) mutational status.
Question 2.
I recommend the authors to check two points: i) lines 104-106 and lines 107-108 say the same thing; ii) lines 108-110 are exactly the same as lines 306-308 in Methods section
Moreover, I suggest removing Paragraph 4.5 “Kaplan-Meier analysis” in Methods section and adding it inside of the following Paragraph 4.6 “Statistical analyses,
Answer 2
Lines 107-108 and lines 306-308 have been erased in the revised version of the manuscript. Paragraph “Kaplan-Meier analysis” has been removed and added in the paragraph “Statistical analyses” (lines 458-460).
Question 3
To improve readability of Table 2, I suggest aligning all the titles for time points, from T2 to T5, on the left.
Answer 3
In Table 2, all the titles for time point T1, T2, T3, T4 and T5 have been aligned on the left.
Question 4
Figure 1 and Figure 2 a-b-c are a little redundant in my opinion. I suggest maintaining only one. In case you prefer to keep both, I invite the authors to combine them in a single multi-panel figure (new Figure 1) but separate from ROC curve (Figure 2).
Answer 4
We agree with the Reviewer that figure 1 and figure 2 a-b-c are redundant. However, to render clearer the results shown in the old figure 1, we prefer to keep both and, as suggested by the Reviewer we combined old figure 2 abc in the new figure 2, panels d, e and f.
Question 5
Finally, minor spell check is recommended. I found some typos throughout the draft, for example: [line 27] Mir, [line 53] such us, [line 119] with low miR-124-3p expression with, [line 295] 16° (missing C).
Answer 5
Spell check was done throughout the manuscript. We thank again the Reviewer for the very careful revision.

Reviewer 2 Report
The study presented by Olioso et al. aims to demonstrate the use of exosomal miRNA expression as predictive biomarkers for high grade gliomas. Authors reported that at least 2 of 3 miRNAs levels are increased in the progressed patients.
The authors need to address the following points:
- Firstly, the scheme of withdrawals is unclear during the article. The study design is only mentioned in the method (fig.4) but it should be moved at beginning of results.
- Little errors in tables are present: i) table 1 numbers of patients in the histology section are shifted ii) and T1 is not aligned in table 2.
- It is not well explained why those miRNAs are chosen, neither in the ref 10 which should support this choice.
- It is not well indicated the threshold limit to determine low or high expression of the studied miRNAs.
- Table 1 presented percentages of molecular features like MGMT methylation and IDH mutation, along with the extension of the resection, but any correlation is done among miRNA expression, these characteristics and PFS or OS. It should be interesting to evaluate this aspect.
- How do the authors explain the inverted expression of all the miRNAs (related to PFS ratio) during the first 10 weeks of treatment (T1-T2)? How can they explain the miR-222 negative prognostic expression only in T5? How do they explain the opposite trend of miR124 at T1?
- miR21 is the most convincible of the listed miRNAs.
- It is not clear if those miRNAs can be used as predictive for the response of therapy or they are simply upregulated when tumor is already progressed. Authors could be clearer in this aspect.
- Have the authors compared serum exosomal miRNAs with total miRNAs? Are there some differences?
- Often those miRNAs are defined as circulating, it is a bit ambiguous and too generic. The term “circulating” should be replaced by exosomal.
- How do miRNAs levels change among patients? How do miRNAs levels change in the same patient according to his therapy response? This could be important for the clinical evaluation of the patients.
- Are the authors sure about the exosomal miRNAs normalization through U6 snRNA? How much abundant is U6 in exosomes? Do the authors tested also the Spike in RNA for normalization?
Author Response
MS “Serum exosomal microRNA-21, 222 and 124-3p as noninvasive predictive biomarkers in newly diagnosed high-grade gliomas: a prospective study”
Reply to Reviewer #2
We are very grateful to the Reviewer for the careful revision and useful comments. The manuscript has been revised according to the specific comments of the Reviewer. With our response to the comments, we hope we have been able to render our findings clearer and our paper acceptable for publication on Cancers.
Question 1.
Firstly, the scheme of withdrawals is unclear during the article. The study design is only mentioned in the method (fig.4) but it should be moved at beginning of results
Answer 1.
To fulfill this important requirement of the Reviewer we added a complete description of the study and moved it together with figure 4 (now figure 1) at the beginning of results (lines 109-119).
Question 2.
Little errors in tables are present: i) table 1 numbers of patients in the histology section are shifted ii) and T1 is not aligned in table 2
Answer 2
Table 1 has been correctly formatted in this revised version.
Question 3
It is not well explained why those miRNAs are chosen, neither in the ref 10 which should support this choice
Answer 3
Rapid development of gene chip technology has offered great help for identification of specific circulating miRNAs. Based on already published data, we selected three miRNAs: miR-21, 222 and 124-3p and focused our attention on this small signature. We wish to apologize for not explaining why these miRNAs were chosen. In the revised version of the manuscript explanation of our choice has been added (lines 75-80).
Question 4
It is not well indicated the threshold limit to determine low or high expression of the studied miRNAs.
Answer 4
As no reference or cut-off value is available for serum exosomal miRNAs, we arbitrarily considered median expression value for separating patients in two groups: “High” and “Low”. Patients displaying miRNA expression above the median were assigned to the subgroup “High” and patients with miRNA expression below the median to the subgroup “Low”. This same criterion allowed the identification of a signature associated with prolonged survival in glioblastoma patients treated with regorafenib (Santangelo A et al. Neuro Oncol. 2021, 23(2): 264-276). We hope to have clarified this point in the revised version of the manuscript (lines 136-137, 458-460).
Question 5
Table 1 presented percentages of molecular features like MGMT methylation and IDH mutation, along with the extension of the resection, but any correlation is done among miRNA expression, these characteristics and PFS or OS. It should be interesting to evaluate this aspect.
Answer 5
Correlation between MGMT methylation, IDH mutation, along with the extent of the resection and miRNA expression is very interesting, as pointed out by the Reviewer. We evaluated any possible relationship between exosomal miRNA expression and MGMT methylation or IDH mutation or the extent of resection. In our cohort of patients, no significant correlation was found. However, since lack of statistically significant correlation could be influenced but the number of subjects included, we did not dare to state the lack of correlation in the manuscript text, as we think that the lack of correlation should be confirmed in a wider group of patients. Therefore, we decided to not include these results in the manuscript.
Question 6
How do the authors explain the inverted expression of all the miRNAs (related to PFS ratio) during the first 10 weeks of treatment (T1-T2)? How can they explain the miR-222 negative prognostic expression only in T5? How do they explain the opposite trend of miR124 at T1?
Answer 6
The inverted expression of all the miRNAs, related to PFS ratio at T1 and T2 time points is not statistically significant. Serum exosomal miRNA expression depends on many different factors, such as, for example, the wide tumor genetic heterogeneity of GBM or the extent of surgical ablation. This can explain the variability of miRNA expression observed in our groups of patients. However, as also shown in the new figure 2, panels A, B, D and E, at T1 and T2 time points no differences were found between stable and progressed patients in miR-21 and miR-222 expression. At T1, miR-124-3p expression is significantly lower in progressed patients than in stable. As already pointed out in the manuscript (lines 338-340), this unexpected finding is difficult to explain. As we have no information regarding pre-operative miR-124-3p expression, we cannot exclude that this subgroup of patients was characterized by a very high miR-124-3p expression before surgery. Interestingly, miR-124-3p expression decreased in stable patients during treatment. As correctly underlined by the Reviewer, miR-222 expression was significantly higher in progressed patients than in stable, only at T5. In our opinion, the variability of miR-222 expression within patients allows to appreciate significant changes only at this time point, when the different expression in the two groups was clearer cut. Although miR-222 alone was a prognostic marker only at the end of the study, the combination with miR-21 and miR-124-3p predicted prognosis at the earlier time point T3.
Question 7
miR21 is the most convincible of the listed miRNAs
Answer 7
We agree with the Reviewer about miR-21, one of the most studied epigenetic biomarkers for glioblastoma. The clinical relevance of miR-21 as a predictive biomarker has already been stressed in the “Discussion” section of the manuscript (lines 345-357).
Question 8
It is not clear if those miRNAs can be used as predictive for the response of therapy or they are simply upregulated when tumor is already progressed. Authors could be clearer in this aspect
Answer 8
We thank the Reviewer also for this very important question. Diagnosis of relapse during post-surgery monitoring of patients with HGG is often challenging. Upregulation of exosomal miRNAs associated with progression may facilitate therapeutic decisions. We think that our study, showing increased miRNA expression during post-operative follow-up of HGG patients, provides useful information about therapy response. Moreover, data shown in the new figure 4 of the revised manuscript indicate that our small signature of exosomal miRNAs can predict progression. As a matter of fact, after RT and first TMZ cycles (T3) patients with no miRNA increased had 33 % probability of relapse within T5 whereas with three miRNAs the probability of recurrence was 89%, even before radiological evidence.
Question 9
Have the authors compared serum exosomal miRNAs with total miRNAs? Are there some differences?
Answer 9
We did not measure the expression of total serum miRNAs, although in the setup of our previous paper (Santangelo et al J Neurooncol 2018) total serum miRNAs were much less informative than the exosomal fraction of serum miRNAs, thus we abandoned the analyses of total serum miRNAs.
Question 10
Often those miRNAs are defined as circulating, it is a bit ambiguous and too generic. The term “circulating” should be replaced by exosomal.
Answer 10
We have replaced circulating with exosomal in the revised version of the manuscript.
Question 11
How do miRNAs levels change among patients? How do miRNAs levels change in the same patient according to his therapy response? This could be important for the clinical evaluation of the patients
Answer 11
Range of miRNA levels among patients can be evaluated by data shown in the new figure 2, panels A, B and C. As already pointed out in answer 6, exosomal miRNA expression is related to many factors. So different patients may have very different miRNA values. Very importantly, miRNA expression may increase or decrease in the same patient and any variation can be clinically relevant.
Question 12
Are the authors sure about the exosomal miRNAs normalization through U6 snRNA? How much abundant is U6 in exosomes? Do the authors tested also the Spike in RNA for normalization?
Answer 12
Before proceeding with the normalization, we checked the stability of U6 snRNA in our cohort of patients. The U6 mean Cq values across time in “Stable” and “Progressed” patients were consistent between the groups and not statistically different. The intra (by time) and inter (by groups) variability calculated as CV% never exceeded 4% value. The abundance of U6 was supported by mean Cq value settled around 25. We did not perform the normalization with Spike-in control because, in our opinion, Spike-in is a good tool in monitoring technical issues, such as RNA extraction and RT reaction efficiency. For normalization we preferred to make use of invariant housekeeping gene that takes into account the relative abundance of miRNAs in exosomal enriched sample.

Reviewer 3 Report
My major comments are about the figures. They are small and the size of the circles on the graphs make it difficult to distinguish the distinct conditions. They should either use different colors or make them circles, triangles, squares or something more distinctive. The lines are difficult to distinguish too.
For Table 2 please define what T1, T2, etc. is in the legend.
There is a typo on line 236. The "m" should not be capitalized.
I would recommend that they switch Figure 4 to Figure 1. It would help set up a better understanding of all the subsequent figures.
Author Response
MS “Serum exosomal microRNA-21, 222 and 124-3p as noninvasive predictive biomarkers in newly diagnosed high-grade gliomas: a prospective study ”
Reply to Reviewer #3
We wish to thank the Reviewer for revising the manuscript and for her/his very useful suggestions to improve the overall presentation of the results.
Question 1.
My major comments are about the figures. They are small and the size of the circles on the graphs make it difficult to distinguish the distinct conditions. They should either use different colors or make them circles, triangles, squares or something more distinctive. The lines are difficult to distinguish too
Answer 1.
Symbols (type and colors) on the figures have been changed according to the reviewer request. We hope that we have been able to improve the quality of the presentation of our results.
Question 2.
For Table 2 please define what T1, T2, etc. is in the legend
Answer 2
T1 up to T5 are the time points of blood sampling as described in the study design, as added in the legend of table 2 of the revised version of the manuscript.
Question 3
There is a typo on line 236. The "m" should not be capitalized.
Answer 3
It has been corrected
Question 4
I would recommend that they switch Figure 4 to Figure 1. It would help set up a better understanding of all the subsequent figures.
Answer 4
Figure 4 is the new figure 1 in the revised version of the manuscript.

Round 2
Reviewer 2 Report
The authors have now answered to most of the concerns raised.